# Social harmony at work: A sharedness index linking team atmosphere to individual well-being in a Japanese company

**Kazuhiro Ito** [1], **Shoko Wakamiya**[1], **Masae Manabe**[2], **Yasushi Watanabe**[2], **Masataka Nakayama**[2], **Yukiko Uchida** [2], **Eiji Aramaki** [1]*

**1** Nara Institute of Science and Technology, Ikoma, Nara, Japan, **2** Kyoto University, Kyoto, Kyoto, Japan

* aramaki@is.naist.jp

## Abstract

Social harmony, defined as the flourishing of interdependent relationships, is central to well-being in workplace teams but is often measured with abstract self-reports that offer limited interpretability and limited guidance for interventions. This study aims to operationalize social harmony as an interpretable sharedness index and to test its association with individual well-being. In a two-month field study at a Japanese company, 94 employees from 23 teams submitted daily reports containing an 11-point well-being score, an 11-point team atmosphere score, and a short diary entry. We computed two weekly team level indices. The score-based sharedness index was defined as the negative of the within team standard deviation of team atmosphere scores. The text-based sharedness index was defined as the mean semantic similarity among members weekly concatenated diaries using Word Mover's Distance. Across team weeks, score-based sharedness correlated positively with mean individual well-being (r = 0.332, p = 0.0002), and text-based sharedness showed a weaker but significant correlation (r = 0.257, p = 0.003). Convergent validity with social harmony was indicated by team level associations with the Interdependent Happiness Scale (score-based r = 0.66, p = 0.019; text-based r = 0.44, p = 0.086). These findings suggest that sharedness reflects a practically interpretable component of social harmony in workplace teams and is positively associated with individual well-being, although causal and intervention effects remain to be tested. The index can be computed from routine ratings and diaries, which supports its use for monitoring and for designing interventions to promote employee well-being.

## Introduction

Well-being encompasses various aspects of life, including the development of one's potential, personal agency, sense of purpose, and positive relationships [1]. It is often considered a more reliable social success than statistical indices, such as economic

**Data availability statement:** We deposit the minimal anonymized dataset and analysis code at GitHub (https://github.com/sociocom/sharedness). Raw text data that might contain personal information cannot be made publicly available under our ethics approval. We therefore release statistically processed data, such as indicator scores, which are sufficient to reproduce all reported results. The dataset reproduces all reported analyses/figures. • Team-by-week table (SSI, TSI, team-week mean of individual well-being) • Team-level table (mean IHS per team) • Emotion table • Odds ratio table • All analysis code (versions pinned).

**Funding:** This work was supported by JST-Mirai Program Grant Number JPMJMI21J2, Japan. The funders had no role in study design, data collection and analysis, decision to publish, or preparation of the manuscript.

**Competing interests:** The authors have declared that no competing interests exist.

indices [2]. The World Health Organization (WHO) defines it as being closely associated with positive mental health [3], and higher well-being has been associated with favorable outcomes in physical health and longevity [4], performance at work [5], and economic wealth [6,7].

In parallel with this scholarly interest, policy and standards have shifted toward workplace well-being over roughly the past decade. National measurement frameworks moved beyond GDP, for example the OECD's How's Life? indicators that mainstreamed subjective well-being in official statistics [8]. Organizational guidance has expanded, with ISO 45003 published in 2021 on managing psychosocial risks [9] and the WHO and the International Labour Organization (ILO) issuing joint guidelines on mental health at work in 2022 [10]. In Asia, policy uptake is visible. Japan's national Stress Check System was introduced in response to growing concerns about work-related mental disorders and karōshi (death from overwork). Since 2015, the system has legally required workplaces with 50 or more employees to conduct annual screening of psychosocial stressors. The program is intended not only to provide workers with individual feedback but also to generate workplace-level data that can guide organizational improvements [11,12]. Singapore's Tripartite Advisory on Mental Well-Being at Workplaces emerged from concerns about sustaining productivity and inclusiveness in the context of changing work arrangements, such as remote work, and an ageing workforce. It was jointly issued by government, employers, and unions, and it promotes organization-level practices including flexible work arrangements, anti-stigma policies, and peer support systems, all designed to safeguard psychological health and foster social integration at work [13]. Taken together, these developments document a sustained, and still expanding, policy and practice emphasis on workplace well-being in both global and Asian contexts.

A long-standing perspective distinguishes affective and cognitive components of well-being. On the affective side, a higher frequency of positive emotions and a lower frequency of negative emotions are thought to contribute to well-being; the Positive and Negative Affect Schedule (PANAS) is a widely used instrument to assess this dimension [14]. On the cognitive side, two constructs are often highlighted: *life satisfaction*, defined as an overall evaluation of one's life according to self-selected criteria [15], and *social harmony*, referring to the flourishing of interdependent relationships within a social context [16]. Instruments such as the Interdependent Happiness Scale (IHS) [17] and the Harmony in Life Scale [18] have extended this cognitive tradition by emphasizing the social harmony dimension that is particularly relevant in interdependent settings.

In workplaces, well-being is linked to outcomes of clear practical importance, including performance, productivity, and retention [5]. While a substantial body of research has examined affective experiences at work and life satisfaction in occupational contexts, social harmony within work teams remains comparatively underexplored. Existing social harmony scales are valuable for broad assessment, yet their items are relatively abstract, which can make interpretation and day-to-day intervention at the team level challenging. There is a need for indicators that preserve conceptual alignment with social harmony while improving interpretability and actionability in organizational practice.

Building on shared reality theory [19,20], we conceptualize *sharedness* as the degree to which team members converge in their assessment of the team's overall psychological state. In our framework, sharedness serves as a concrete, team-level operationalization of social harmony. We emphasize that sharedness does not imply that members have similar levels of individual well-being; rather, it reflects similarity in members' evaluations of the team atmosphere as a whole, that is, a shared appraisal of how the team is doing overall on a given day (Fig 1). Conceptually, higher sharedness may reduce ambiguity about the team's situation, facilitate coordination of expectations, and support prosocial responses among coworkers, which in turn could relate to higher individual well-being. This rationale is especially pertinent in Japanese companies, where interdependent coordination and consensus-oriented practices can make a shared understanding of the team's state particularly consequential for day-to-day functioning.

Among various possible team psychological states related to sharedness, this study employs the following two different indices:

- Score-based sharedness: The agreement of team atmosphere score
- Text-based sharedness: The agreement of diary contents

For qualitative sharedness, the diary text was analysed using Natural Language Processing (NLP). This qualitative index has the advantage of flexibility, as it can be calculated using only textual data, allowing for more nuanced insights into team dynamics without the need for extensive quantitative measures. These indices are designed to be computable from routinely collected ratings and diaries, easy to interpret, and amenable to intervention in the workplace compared to traditional measures.

Within this framework, we examine the relationships among sharedness, social harmony, and individual well-being in Japanese work teams. We state three hypotheses. First, higher score-based sharedness will be positively associated with the weekly mean of individual well-being across team members. Second, higher text-based sharedness will likewise be positively associated with the weekly mean of individual well-being. Third, both indices will show positive correlations with an established social harmony measure (IHS) at the team level. These expectations are grounded in the interdependent coordination that characterizes many Japanese companies, where shared understanding is often integral to effective collaboration.

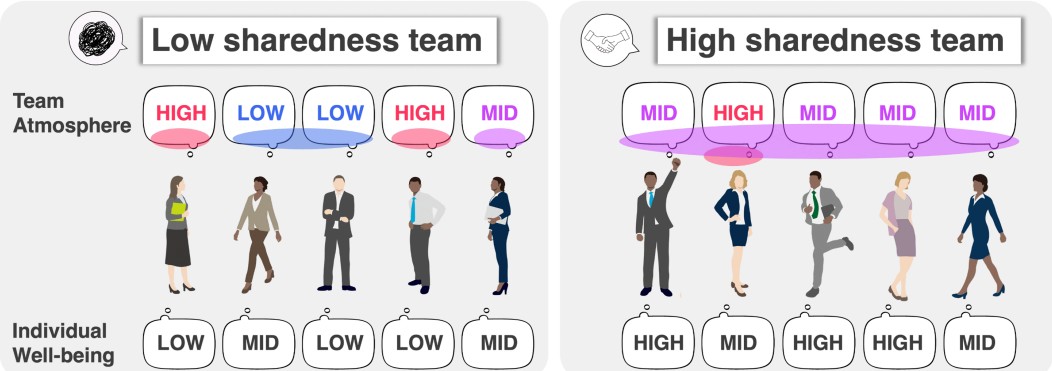

**Fig 1**. **Concept overview.** The top row shows members' ratings of team atmosphere, and the bottom row shows individual well-being. The left team has inconsistent atmosphere ratings (low sharedness), whereas the right team has consistent ratings (high sharedness). We hypothesize that greater sharedness is associated with higher individual well-being.

In positioning this study, we explicitly compare our approach with systematic, survey-based approaches that are widely used to assess well-being and social harmony (e.g., the IHS and Harmony in Life). Systematic approaches rely on standardized items administered at planned intervals and offer strong psychometric grounding and comparability across groups. By contrast, our sharedness indices are designed to be team-anchored and high-frequency: the score-based index captures within-team agreement in atmosphere ratings, and the text-based index captures semantic convergence in routine diaries. This design aims to complement standardized scales in four ways: (i) unit of analysis (team-level convergence rather than only individual attitudes), (ii) temporal granularity (weekly summaries from daily inputs rather than infrequent cross-sections), (iii) interpretability for interventions (direct signals of team consensus or fragmentation), and (iv) respondent burden (using existing diaries with minimal additional load). We do not seek to replace validated instruments; instead, we assess convergent validity by relating our indices to IHS at the team level and examine their association with weekly mean individual well-being in Japanese work teams. Operational definitions and analytic procedures are detailed in the Materials and Data Analysis sections. Our rationale is pragmatic: to preserve the conceptual core of social harmony while offering an interpretable, low-burden, team-level indicator that can inform timely workplace actions.

## Materials and methods

### Participants

We collected daily reports from employees at the Kansai office of Hakuhodo Inc. (https://www.hakuhodo-global.com/network/japan/), a major advertising and marketing company in Japan, for two months between 1 September and 31 October 2022. The statistical population for this study comprised all employees at the Kansai office during the study period; all such employees were eligible to participate. A total of 121 employees provided informed consent in advance, and 94 of them submitted at least one daily report during the two months. Employees belonged to one of 23 teams; each team was identified by a letter from A to W. Each team comprised four to seven members, and the number of members who submitted one or more daily reports ranged from one to seven per team. The average age of participants was 39.4 years, and 27.5% were female. As recruitment was limited to a single site and industry, generalizability beyond this population should be interpreted with caution.

**Participant consent.** This research was approved by Kyoto University, including Nara Institute of Science and Technology (Review No. 26-P-16). All methods were performed in accordance with relevant guidelines and regulations. Before beginning the questionnaire, participants read an online explanation of the study overview and proceeded only after providing consent.

### Questionnaire

To collect daily reports, we developed a web-based form using Streamlit (https://streamlit.io/). The form had three input fields: diary entry, individual well-being score, and team atmosphere score (Table 1). The diary entry was free text; content did not have to be related to the company or work. Individual well-being and team atmosphere were each rated on an 11-point scale ranging from 0 (very negative) to 10 (very positive). Note that although the form instructions and the diaries entered by participants were in Japanese, English translations are shown in this paper.

In addition, to validate our proposed sharedness index, participants answered the Interdependent Happiness Scale (IHS), an existing measure of social harmony, after the experimental period. For IHS, participants responded to all nine items (e.g., *You believe that you and those around you are happy*). See Supporting information S1 Table for the full item list.

**Table 1**. **Overview of the web-based daily report form.**

| Item | Question | Format |
|---|---|---|
| Diary entry | Fill in your diary with about three lines. Note that it is acceptable to include content unrelated to work. | Free text with an unlimited number of characters; required |
| Individual well-being score | Did you feel a sense of well-being today? | 11-point scale from 0 (very low well-being) to 10 (very high well-being); required |
| Team atmosphere score | Do you think the team was in a state of well-being today? | 11-point scale from 0 (very negative) to 10 (very positive); required |

## Instructions

At least once a day, typically at the end of the workday (optional on non-work days), participants logged into the form from their own devices and filled out a daily report. If more than one report was entered on the same day, diary texts were concatenated, individual well-being and team atmosphere scores were averaged. After submitting, participants received feedback on their own individual well-being scores (for the past seven days) and the distribution of individual well-being scores from other members of the same team. Fig 2 shows screenshots of the login, input, and feedback pages translated from Japanese to English.

The following data sources were examined to study the relationships among the variables of interest:

**Diary entry:** Participants were instructed to write a short free-form diary of about three lines; we did not instruct them to include content related to well-being or emotion. For example, *"A meeting was held in the evening. The meeting was a great opportunity to hear the opinions of members who do not usually talk with me, so it was a meeting with*

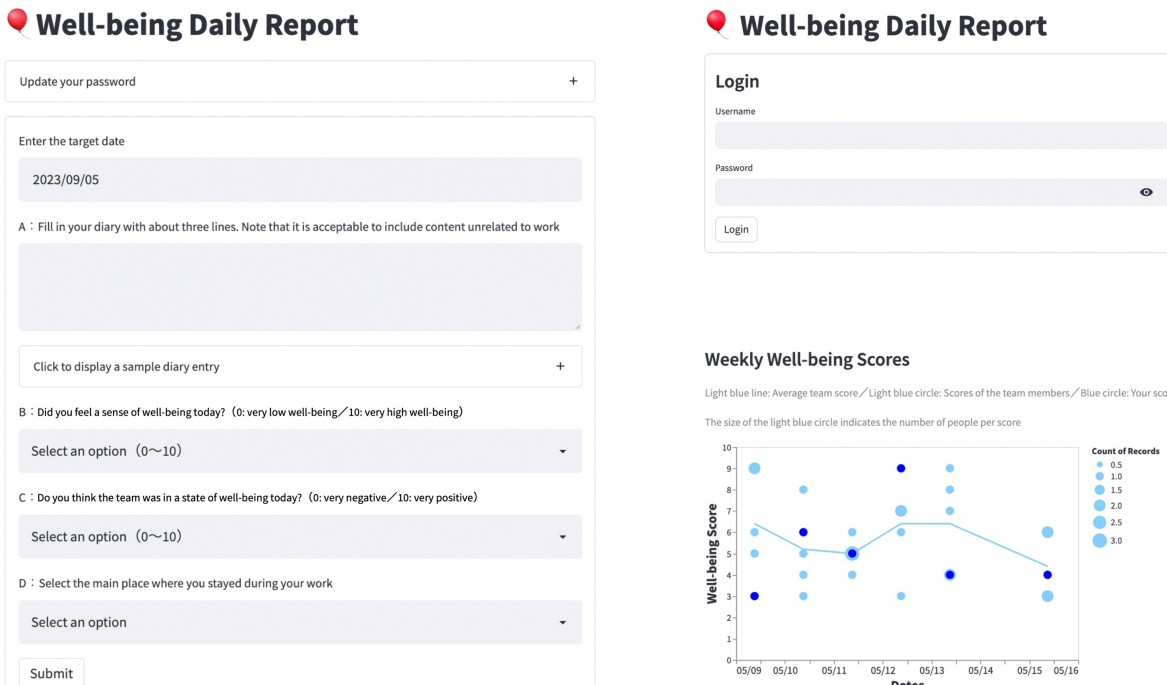

**Fig 2**. **Screenshots of the web-based daily report system translated from Japanese to English.** Participants first log in from the login page (upper right) and enter their daily reports on the input page (left). After the submit button is pressed, the feedback page (lower right) appears at the bottom of the page.

*many discoveries."* and *"On my way home from work, I found a restaurant that I really like. I want to take someone there one of these days."* Participants were asked not to include personal information or company-confidential information.

**Individual well-being score:** Self-reported on an 11-point scale from 0 (very low well-being) to 10 (very high well-being).

**Team atmosphere score:** The participant's assessment of the team's atmosphere on an 11-point scale from 0 (very negative) to 10 (very positive).

## Data analysis

**Analytical framework.** The analytical unit was the team by week. Weeks were defined as consecutive seven-day periods anchored at 2022-09-01; Week 1 began on 2022-09-01 and covered the period from 2022-09-01 to 2022-09-07. Aggregations honored these fixed seven-day windows. For each team and week, we aggregated daily inputs to weekly summaries and constructed two team-level sharedness indices (score-based and text-based) that operationalize social harmony. We then examined their associations with the weekly mean of individual well-being across team members. An overview of the workflow is shown in Fig 3.

This was a non-interventional observational study with daily self-reports over nine weeks (from 2022-09-01 to 2022-10-31). Team-level sharedness indices (SSI, TSI) were computed weekly, and the Interdependent Happiness Scale (IHS) was administered once after the reporting period. No randomization or experimental manipulation was performed.

**Score-based sharedness index.** The score-based sharedness index ($SSI_{t,w}$) was defined as the negation of the within-team standard deviation of team atmosphere scores, averaged across days within week $w$, as follows:

$$SSI_{t,w} = -\,\mathrm{mean}(\sigma_1, \sigma_2, \ldots, \sigma_n),$$

where $t \in T$ is a specific team, $w \in W$ is a specific week, $\sigma_i$ is the standard deviation of team atmosphere scores reported by members of team $t$ on day $i$ of week $w$, and mean($\cdot$) denotes the average across days with inputs from two or more members. The procedure for measuring the score-based sharedness index for each team is shown in Fig 3a. We examined Pearson correlation coefficients between the score-based sharedness index and the average individual well-being. Days with inputs from fewer than two participants were excluded.

Next, we examined Pearson correlations between the score-based sharedness index and average individual well-being for each team using the same weekly aggregation. Only dates with inputs from at least two members were used; teams contributing four or more weekly observations (approximately half of the study period) were included in the per-team analysis.

Finally, to validate the proposed index, we computed the correlation between the score-based sharedness index and IHS, which assesses subjective well-being in terms of harmony with others and collective relations [17], at the team level. For the score-based sharedness index, values were averaged across all weeks per team. Teams with fewer than three weekly values were excluded. For IHS, the nine items were averaged to obtain an individual score and then averaged within teams. In psychological research, a correlation coefficient of 0.4 or higher is often taken to indicate that one indicator can serve as a surrogate for another [21]; we adopt this criterion here.

**Text-based sharedness index.** The text-based sharedness index ($TSI_{t,w}$) was defined by the semantic similarity of diaries among team members as follows:

$$TSI_{t,w} = \frac{1}{C} \sum_{\substack{i,j \in t \\ (i \neq j)}} \mathrm{sim}(v_{i,w}, v_{j,w}),$$

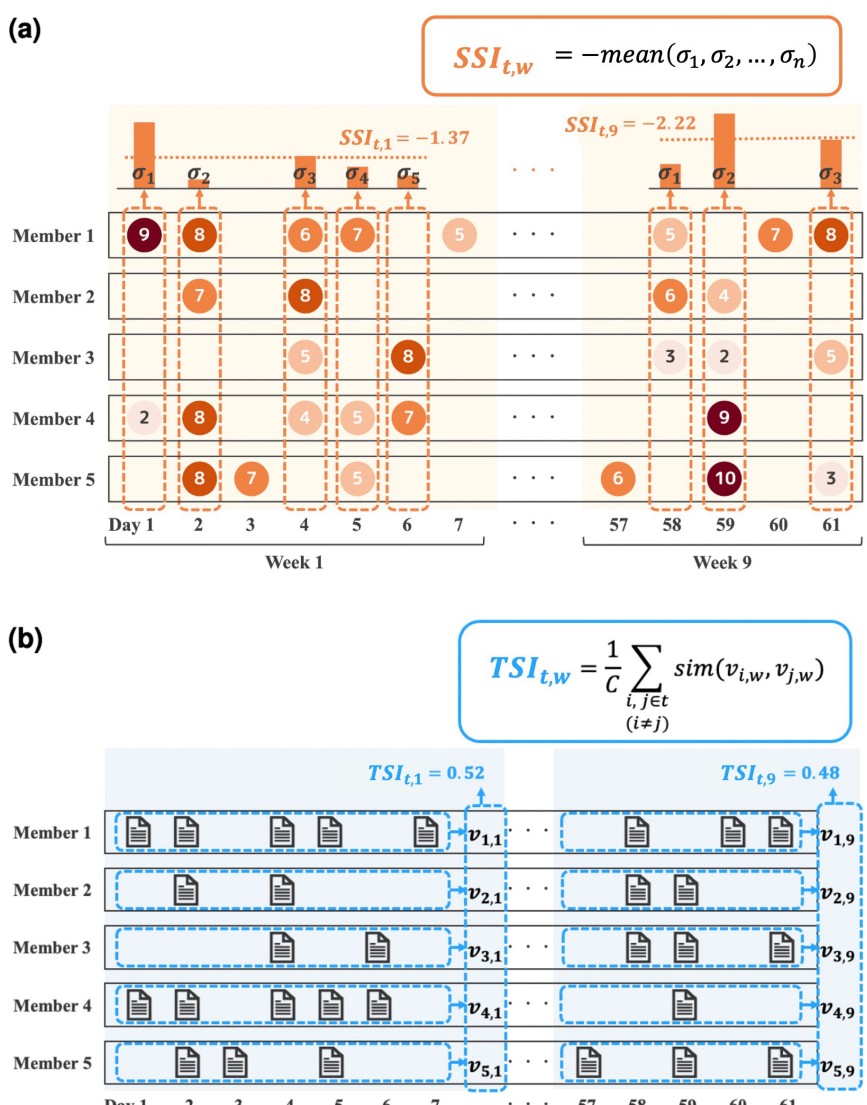

**(a)**

$$SSI_{t,w} = -mean(\sigma_1, \sigma_2, ..., \sigma_n)$$

**(b)**

$$TSI_{t,w} = \frac{1}{C} \sum_{\substack{i,j \in t \\ (i \neq j)}} sim(v_{i,w}, v_{j,w})$$

**Fig 3. Overview of index construction.** (a) SSI: within-team daily SD of atmosphere (sign-reversed), averaged within week. (b) TSI: mean pairwise semantic similarity among weekly concatenated diaries.

where $t \in T$ is a specific team, $w \in W$ is a specific week, $C$ is the number of unique member pairs in team $t$, and $v_{i,w}$ and $v_{j,w}$ are vector representations of the weekly concatenated diary texts for members $i$ and $j$, respectively. Here, $sim(\cdot, \cdot)$ denotes a similarity transformed from a semantic distance. Semantic distances were computed using Word Mover's Distance (WMD) [22] with the Japanese Wikipedia Entity Vectors pre-trained Word2Vec model (300-dimensional; `jawiki.all_vectors.300d.txt.bz2`) [23,24]. Tokenization for Japanese texts followed MeCab [25] using the NEologd [26] dictionary. Additional implementation details are provided in the Supporting information S2 Algorithm. The procedure used to measure the text-based sharedness index for each team is illustrated in Fig 3b.

We provide a technical supplement on the text-based sharedness index. In this index, we employ the WMD to assess the semantic similarity between sentences. Unlike simple edit distance, which measures the minimum number of operations required to transform one sentence into another, WMD captures the meaning of the sentences by considering the contextual relationships between words. For instance, consider the following sentences:

- *The manager praised the team for their excellent work.*
- *The manager scolded the team for their poor performance.*

From an edit distance perspective, these sentences might be considered similar because they share several common words and only require a few word substitutions. However, semantically, these sentences convey opposite meanings— one is positive, and the other is negative. As a result, their WMD score would indicate low similarity, highlighting the distinction between surface-level text similarity and deeper semantic similarity. This example illustrates how WMD is more suited for capturing the true meaning of sentences.

We examined Pearson correlations between the text-based sharedness index and the average individual well-being. As with the score-based index, we also examined Pearson correlations within teams using the same weekly aggregation. Only dates with inputs from at least two employees were used, and only teams with four or more weekly observations (approximately half of the study period) were included in the per-team analysis.

Finally, to validate the proposed index, we computed the correlation between the text-based sharedness index and IHS at the team level. For the text-based index, values were averaged across all weeks per team. Teams with fewer than three weekly values were excluded. For IHS, the nine items were averaged to obtain an individual score and then averaged within teams.

**Linguistic analysis: Words and emotions. Words.** Furthermore, to explore words associated with individual well-being and the sharedness index, we analyzed word-level odds ratios in diaries. This subsection focuses on the score-based sharedness index because it showed the stronger correlation with individual well-being and does not depend on text modeling. We constructed two classes for each target: for individual well-being, a high-score class (third quartile or higher; scores $\geq 8.0$; 803 reports) and a low-score class (first quartile or lower; scores $\leq 5.0$; 477 reports); for the sharedness index, a high class (score-based index $\geq -0.70$; 53 reports) and a low class (score-based index $\leq -2.18$; 108 reports).

MeCab [25] with the NEologd dictionary [26] was used for tokenization. Note that because the analysis unit is a word, some Japanese words appear as multi-word phrases after translation to English.

**Emotions.** We expected that positive emotions inferred from diaries would be positively correlated with individual well-being, whereas negative emotions would be negatively correlated [27,28]. If supported, this would indicate that diary texts are consistent with reported well-being and that texts can capture broader psychological states. To estimate emotions, we utilized the LIFE STORY corpus (https://sociocom.naist.jp/life-story-data/). LIFE STORY is an open corpus of episodes associated with seven emotions based on Plutchik's theory ("sadness," "anxiety," "anger," "disgust," "trust," "surprise," and "fun") [29], collected via crowdsourcing four times a year. Each wave involves 1,000 participants contributing one text per emotion, yielding 7,000 texts. We trained an emotion classification model using data from season 1 of 2017 to season 3 of 2020.

For classification, we fine-tuned the publicly available Japanese BERT model `cl-tohoku/bert-base-japanese-v2` [30,31] using the Hugging Face Transformers library with PyTorch Lightning. Training was performed with a maximum sequence length of 256, batch size of 16, weight decay, and a learning rate in the range of $1 \times 10^{-5}$–$5 \times 10^{-5}$. To address class imbalance, class weights were incorporated into the cross-entropy loss. Early stopping with a patience of 3 epochs was applied based on validation macro F1-score, and the best model was checkpointed automatically. To optimize

hyperparameters, we conducted Optuna-based [32] search over learning rate, weight decay, and warmup ratio. The best-performing configuration was $lr = 2.6\times10^{-5}$, weight decay $= 7.1\times10^{-6}$, and warmup ratio $= 0.036$. On the held-out test set, the model achieved a macro F1-score of 0.72.

## Statistical analysis

Unless noted otherwise, the procedures below apply to both sharedness indices (SSI and TSI). We computed two-sided Pearson correlation coefficients between each sharedness index (SSI, TSI) and the weekly mean of individual well-being across all team by weeks, with $\alpha = 0.05$. We report Pearson correlations $r$ with two sided $p$ values. We then repeated the correlations within teams and within weeks as exploratory analyses; owing to smaller sample sizes, we treat $p < 0.10$ as suggestive evidence and report unadjusted $p$-values. Exclusions and aggregation rules followed the criteria described above. Analyses were conducted in Python 3.10.18 (pandas 2.3.1, scipy 1.15.3); exact versions and executable scripts are available in the project repository (see Data Availability Statement).

## Results

The summary statistics of the daily reports are as follows. The mean number of data inputs for the days was 41.7, and the median was 40.5. Throughout the experimental period, the number of inputs decreased. The average individual well-being score was 6.6 ($sd$=2.1), and the average team atmosphere score was 6.4 ($sd$=1.8).

To examine our hypothesis, that individuals within teams exhibiting greater sharedness report higher levels of individual well-being, we analyzed data from 94 employees and evaluated associations at the team-by-week level using two sharedness indices. The primary analysis used team by week observations, defined as team-by-week aggregates with reports from at least two members; days with fewer than two contributors were excluded. Out of 207 potential team by weeks (23 teams $\times$ 9 weeks), 118 (57%) met SSI inclusion and 133 (64%) met TSI inclusion. In team-stratified analyses, we retained 12 teams for SSI and 16 teams for TSI (each contributing $\geq 4$ team by weeks). The primary results are shown in Table 2.

## Score-based sharedness index

The score-based sharedness index and individual well-being indicated a positive correlation coefficient of 0.332 ($p$=0.0002), as shown in Fig 4a, supporting our hypothesis. Of the 23 teams, we applied a per-team analysis to 12 teams in which two or more members submitted reports for four or more weeks, as shown in Fig 4b and S3 Table in the Supporting Information. Three of these showed positive correlations between the indices (Team G ($r$=0.973, $p$=0.027), Team J ($r$=0.677, $p$=0.045), and Team L ($r$=0.777, $p$=0.023)), similar to the overall results, and Teams G and L exhibited strong positive correlations ($r >= 0.7$). In contrast, only Team A ($r$=-0.75, $p$=0.086) showed a negative correlation.

We also performed the same analysis to test for correlations on a weekly basis as shown in Fig 5a and S4 Table in the Supporting Information. Five of the nine weeks showed significantly positive correlations at $p < 0.10$ (Weeks 3, 4, 7, 8, and 9), while there were no weeks with significantly negative correlations. Similar to the all-week results, the per-week results demonstrated a positive correlation between sharedness index and individual well-being.

**Table 2. Primary associations of sharedness indices with individual well-being and IHS.** Pearson correlations (two-sided). For WB, $N$ is the number of team by weeks; for IHS, $N$ is the number of teams. Abbreviations: WB, individual well-being; IHS, Interdependent Happiness Scale; SSI, score-based sharedness index; TSI, text-based sharedness index.

| | WB (team by week) | | | IHS (team level) | | |
|---|---|---|---|---|---|---|
| | $r$ | $p$ | $N$ | $r$ | $p$ | $N$ |
| Score-based sharedness index | 0.332 | 0.0002 | 118 | 0.662 | 0.019 | 12 |
| Text-based sharedness index | 0.257 | 0.0030 | 133 | 0.443 | 0.086 | 16 |

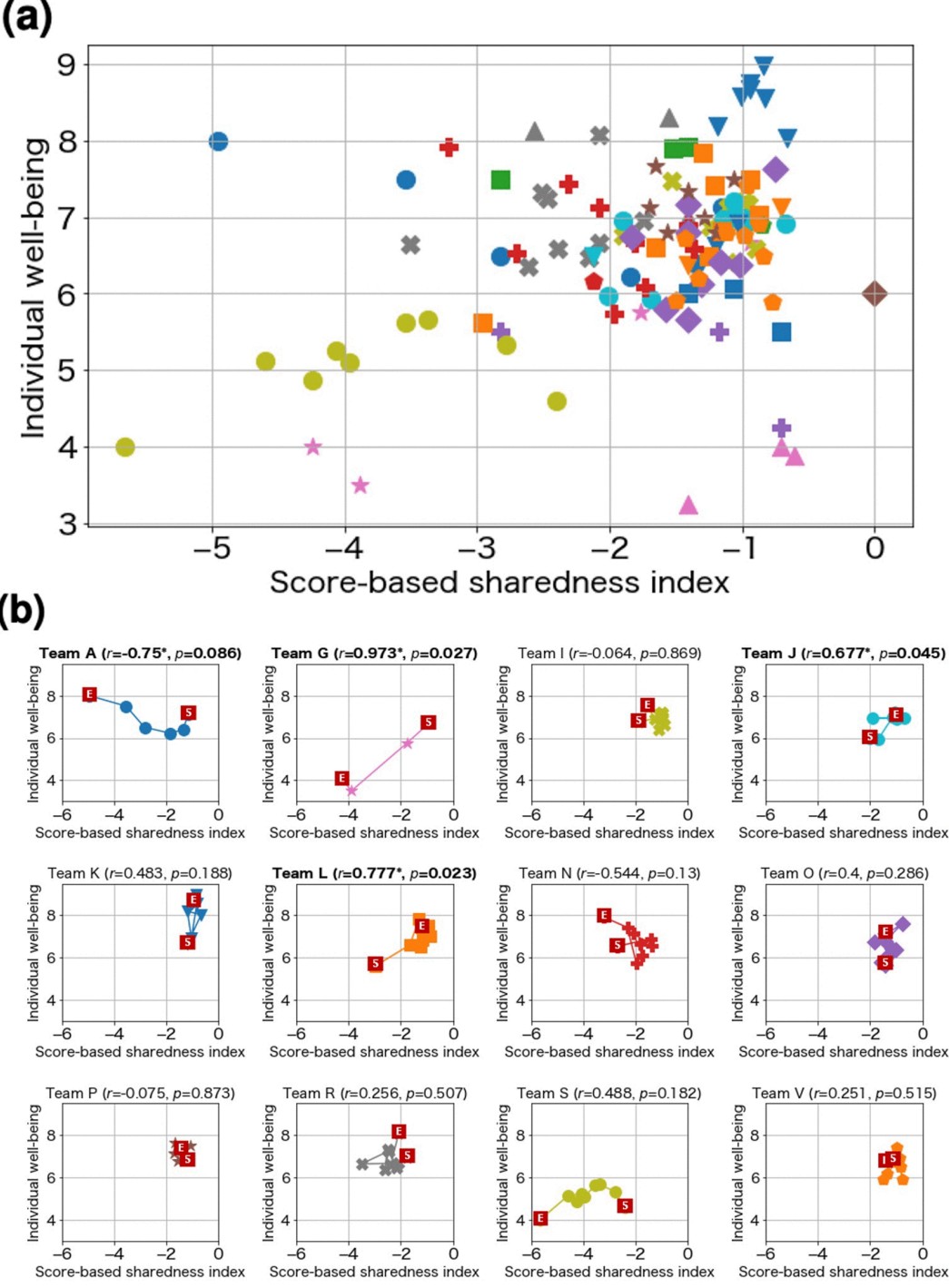

**Fig 4**. **Results for score-based sharedness index.** Each plot shows the values per week and per team. Markers of the same color and shape indicate the same team. (a) Result for all teams. The correlation coefficient is 0.332 ($p$=0.0002), supporting our hypothesis. (b) Results for each team. The "S" indicates the score on the first day, and the "E" indicates the score on the last day. "*" means that the value is statistically significant ($p < 0.1$).

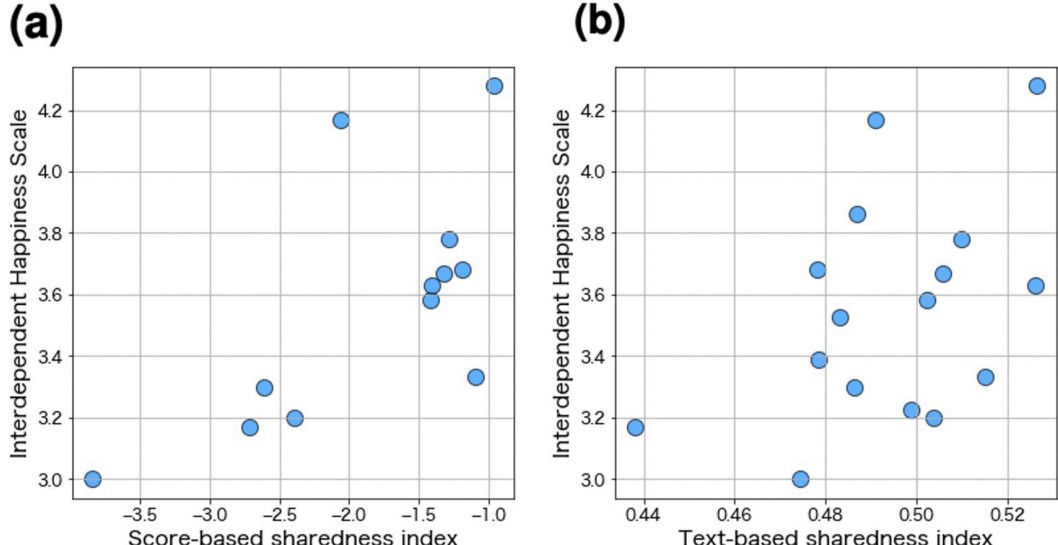

**Fig 5**. **Results divided by week.** "*" means that the value is statistically significant (*p* < 0.1). (a) Score-based sharedness index. (b) Text-based sharedness index.

Finally, the correlation coefficient between the score-based sharedness index and IHS was 0.66 (*p* = 0.019, Fig 6a). Since it is above 0.4, this index works as a surrogate indicator of social harmony.

**Text-based sharedness index**

The text-based sharedness index yielded a correlation coefficient of 0.257 (*p*=0.003) with individual well-being as shown in Fig 7a, similar to the score-based sharedness index scores. The correlation coefficient between text-based sharedness index and score-based sharedness indices was 0.396 (*p*=0.00001), indicating that these two factors show positive correlation. Of the 23 teams, we applied a per-team analysis to 16 teams in which two or more members had submitted reports for four or more weeks, as shown in Fig 7b and S3 Table in the Supporting Information. Only Team O (*r*=0.635, *p*=0.066) showed a significant negative correlation between the indicators, whereas the other teams did not show any significant correlations. The per-team score revealed that unlike the combined results for all teams, no team showed a significant positive correlation. This implies that while there is a positive correlation between the indicators when comparing between teams, the correlation when stratified by team remains unknown, which is a limitation of our experiment.

We also performed the same analysis to test for weekly correlation as shown in Fig 5b and S4 Table in the Supporting Information. Five of the nine weeks showed significantly positive correlations at *p* < 0.10 (Weeks 3, 4, 6, 7, and 8), while there were no weeks with significantly negative correlations. Similar to the all-week results, the per-week results demonstrated a positive correlation between sharedness index and individual well-being. Furthermore, significant positive correlations were found for weeks 3, 4, 7, and 8 using either the score-based sharedness index or text-based sharedness indices, suggesting the consistency of the two sharedness indices.

Note that the threshold value of p-value was set at 0.1 for correlations by team and by week, because the small number of data points made it difficult to obtain significant values in the tests. Therefore, more data should be collected and verified in the future to see if the correlations are consistent when stratified by team or period.

Finally, the correlation coefficient between the text-based sharedness index and IHS was 0.44 (*p* = 0.086, Fig 6b). Since it is above 0.4, this index works as a surrogate indicator of social harmony.

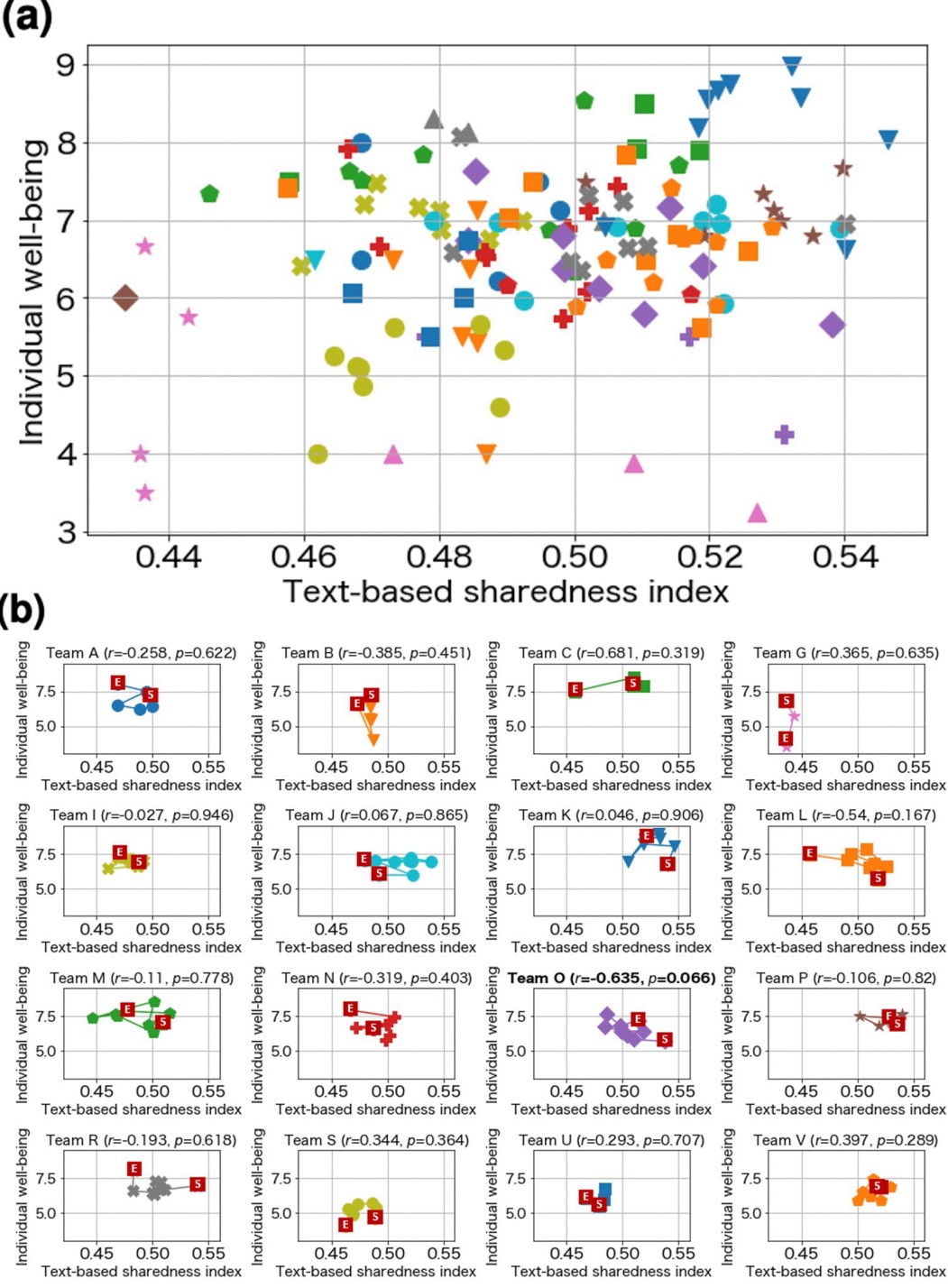

**Fig 6**. **The correlation coefficient between the sharedness indices and Interdependent Happiness Scale (IHS).** Each plot corresponds to each team. Because both correlations exceed 0.4, the indices can serve as surrogate indicators of social harmony.

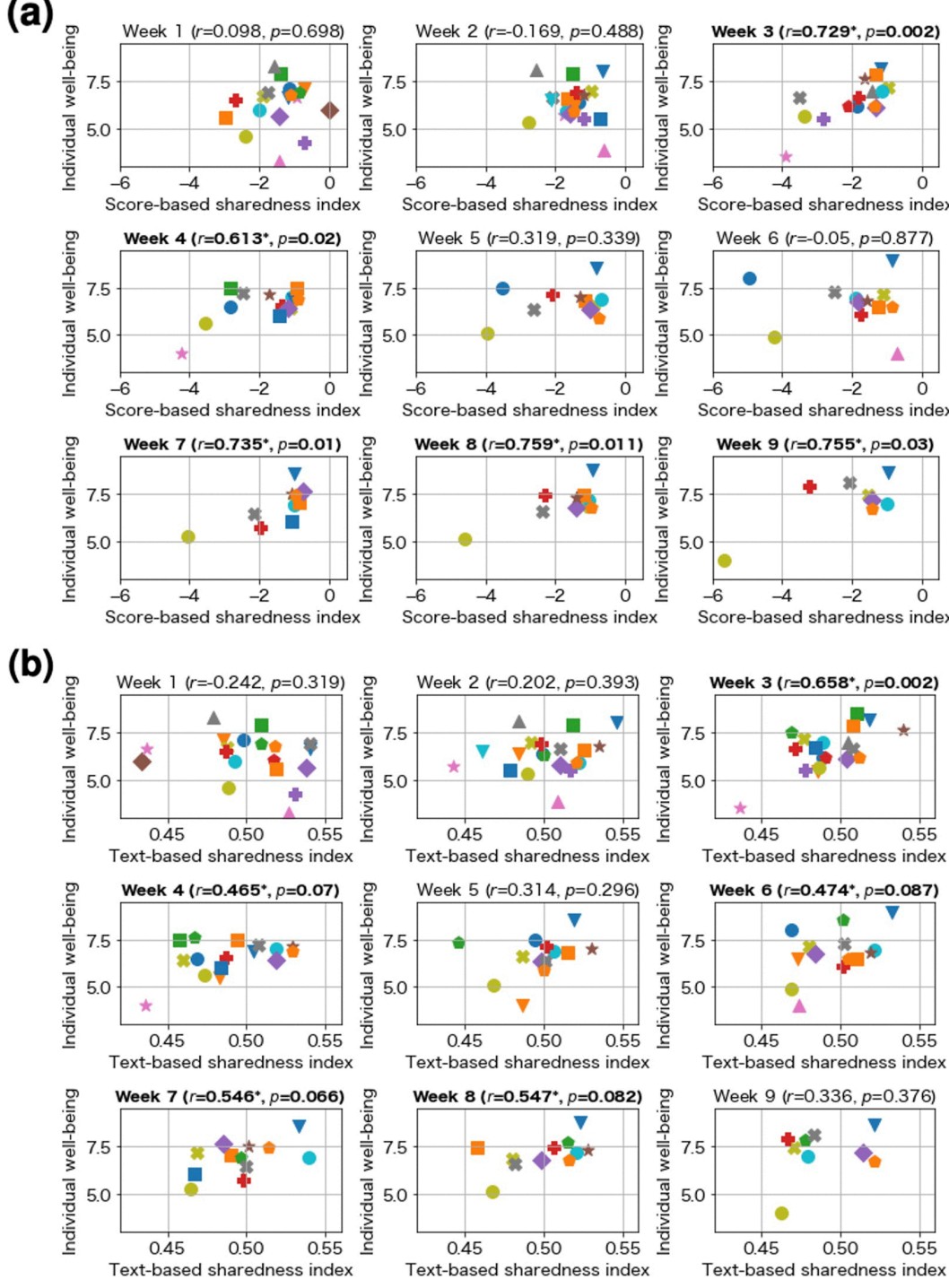

**Fig 7**. **Results for text-based sharedness index.** Markers of the same color and shape indicate the same team. (a) Result for all teams. (b) Results for each team. The "S" mark indicates the score on the first day and the "E" mark the position of the score on the last day. "*" means that the value is statistically significant ($p < 0.1$).

## Linguistic analysis: Words and emotions

**Words.** The upper part of the Table 3 shows the words characteristic of the high and low individual well-being classes. In the high well-being class, words that express positive emotions such as "happiness," and "fun," words that suggest interactions with others such as "group company", "senior member", and "meal", and words that indicate irregular work such as "photoshooting" and "Tokyo" appear. In addition, the lower part of the Table 3 shows words of the high and low score-based sharedness index classes. The high sharedness index class shows words related to interactions with others such as "lunch", while the low sharedness index class shows words for collective people such as "group company" and "within the company". Note that in this part, although the unit of analysis is a word, there are cases where what was a word in Japanese is a phrase of multiple words as a result of translation into English.

**Emotions.** Table 4 summarizes individual well-being by predicted emotion. Relative to the overall mean ($\overline{\text{WB}}$ = 6.802), well-being was lower for Anger, Disgust, Sadness, and Anxiety, and higher for Joy and Trust. Surprise was nearly unchanged. These findings suggest that positive emotions (Joy, Trust) are linked with higher well-being, while negative emotions (Anger, Disgust, Sadness, Anxiety) correspond to lower levels. This supports the possibility of estimating well-being from diary text and capturing broader aspects of psychological state.

**Table 3**. **Words with high and low odds ratios for the high and low individual well-being classes (upper) and those for the high and low score-based sharedness index classes (lower).** Several "words" are composed of multiple words, which is the result of translation from Japanese to English.

| High individual well-being class | | Low individual well-being class | |
|---|---|---|---|
| Word | Odds Ratio | Word | Odds Ratio |
| group company | 15.525 | regret | 0.038 |
| participation | 7.438 | exhausted | 0.043 |
| senior member | 7.263 | mistake | 0.052 |
| happiness | 6.541 | mental | 0.052 |
| laugh | 6.061 | list | 0.052 |
| meal | 5.663 | winter | 0.052 |
| work | 3.868 | hospital | 0.052 |
| meeting | 3.749 | tough | 0.052 |
| happy | 3.616 | my limit | 0.052 |
| fun | 3.514 | positive | 0.052 |
| photoshooting | 3.443 | hate | 0.067 |
| Tokyo | 3.315 | depression | 0.100 |
| **High score-based sharedness index class** | | **Low score-based sharedness index class** | |
| Word | Odds Ratio | Word | Odds Ratio |
| lunch | 20.461 | group company | 0.028 |
| weekdays | 6.798 | participation | 0.185 |
| competitor | 6.798 | meeting | 0.197 |
| delicious | 6.798 | within the company | 0.202 |
| near | 4.378 | response | 0.296 |

**Table 4**. **Individual well-being by predicted emotion (emotion labels are outputs of the LIFE STORY–trained classifier).** $\Delta = \overline{\text{WB}}_{\text{emotion}} - \overline{\text{WB}}_{\text{all}}$; positive values indicate higher well-being than the overall mean, negative values indicate lower.

| Emotion | N | WB Mean | WB SD | Δ vs. overall mean |
|---|---|---|---|---|
| All data | 1,741 | 6.802 | 2.143 | – |
| Joy | 647 | 7.536 | 1.752 | 0.734 |
| Trust | 498 | 7.288 | 1.687 | 0.486 |
| Anxiety | 302 | 5.492 | 2.197 | -1.310 |
| Surprise | 103 | 6.670 | 2.320 | -0.132 |
| Sadness | 72 | 5.778 | 2.445 | -1.024 |
| Anger | 63 | 4.976 | 2.531 | -1.826 |
| Disgust | 56 | 4.679 | 2.124 | -2.123 |

## Discussion

### Principal results

The experimental results revealed a positive relationship between sharedness index and individual well-being. Our hypothesis was supported by both types of sharedness index. A qualitative interpretation of the results is that when members' assessments of a team's psychological state are not consistent within the team, it is difficult for members to behave empathetically with each other. When empathetic behaviour is difficult, communication with inappropriate emotional expressions is more likely to occur [33], which can result in lower individual well-being. Our results are consistent with the theories of shared reality [19,34], the Interdependent Happiness Scale [17], and the Harmony in Life Scale [18] in social psychology. However, the causal relationship between the two concepts (sharedness index and individual well-being) is still unclear; whether sharedness is the cause of well-being or, conversely, whether teams with high individual well-being have high sharedness remains unknown.

From a macro perspective rather than a causal relationship, it is also important to understand the dynamics. In psychology, there are many studies on the dynamics between group and individual emotions. Barsade et al. categorized how group-level emotions are structured into four types: bottom-up processes, which include (a) convergence in group affect and (b) affective diversity, and top-down processes, which encompass (c) emotional culture and (d) dynamic processes that evolve over the lifespan of the group [35]. Within these categories, our hypothesis is closer to the top-down (c) emotional culture, which is described as "*Behavioral norms, artifacts, and underlying values and assumptions reflecting the actual expression or suppression of the discrete emotions comprising the culture and the degree of perceived appropriateness of these emotions, transmitted through feeling and normative mechanisms within a group*". In this context, our study can be explained as the more team members share perceptions about how to express well-being (instead of norms about how to express emotions), the higher their well-being. Thus, from the perspective of emotion culture, it can be inferred that the group state influenced the individual well-being.

On the other hand, Hareli and Rafaeli [36] proposed a model in which individual mood influences the members of a group, and the cycle of influence continues among group members, causing convergence or divergence of mood. As in Hareli and Rafaeli's model, it is quite possible that individual well-being and sharedness are not simply a relationship in which one affects the other, but are cyclical influences in the interaction among group members. In the future, based on psychological theories such as these, we will use longitudinal data to identify dynamics related to the psychological states of groups and individuals.

### Linguistic analysis: Emotions and words

Our results of emotions are consistent with previous studies that have suggested the relationship between linguistic features and individual well-being [37–40]. The results suggest that the collection of diaries can be used to estimate well-being and emotion trends at the same time.

In addition, the word-level analysis examined words that were highly associated with high and low classes for the individual well-being and sharedness index. This suggested events associated with each class. Words that tended to appear in the high individual well-being class included "group company" and "senior member," suggesting that the members were interacting with others who belonged to the same organization. These lexical patterns suggest that positive affect, social contact, and episodic novelty/variety co-occur with higher individual well-being. Also, words such as "lunch" that indicate interactions with others appeared in the high sharedness index class and the low sharedness index class shows words for collective people such as "group company" and "within the company". Together, the terms suggest that informal co-present interactions promote within-team consensus, whereas dispersed organization-level tasks reduce it. The fact that words indicating interactions with others also appeared in the high individual well-being and sharedness index classes suggests that the factor of interactions with others is one of the mediators of the correlation between these values. This

result is consistent with reports that social capital enhances well-being [41] and with theories about well-being in interactions with others [17].

On the other hand, "Group company" and "participation" appear in both the high individual well-being class and the low sharedness class. This suggests an exceptional case in which an event is both a factor that raises individual well-being and a factor that lowers sharedness index, which is contrary to the results of the correlation analysis for the hypotheses. For example, irregular tasks, such as working with a group company, may increase individual well-being but decrease sharedness index because they are not related to team members. Although qualitative analysis is essential to identify such factors, we believe that this kind of investigation is important to elaborate the relationship between individual well-being and team conditions.

Through emotions and word-level analyses of diaries, we have shown that text can capture the well-being of the writer while also suggesting the character of the individual and the organization. To take our study of the correlation between individual well-being and sharedness index further, it is important to capture the characteristics of individuals and organizations that condition the relationship between individuals and groups. We suggest that NLP is useful as a key technology for this purpose.

## Research limitations

This study is the first step in one of the few attempts to investigate the dynamics between group psychological states and individual well-being. Thus, there are many challenges ahead of us, as described below.

### Correlational design and lack of causality

The results support our hypothesis that "individuals within teams exhibiting greater sharedness have higher individual well-being". However, as these results are merely correlations and do not establish causality, it remains unclear whether higher sharedness index is linked to higher levels of individual well-being. Moving forward, we aim to examine the causal relationships.

### Heterogeneity and potential mediators across teams and individuals

In general, although our results support our hypotheses, when considering each team, some teams matched the hypotheses, whereas others did not. This implies diversity in the influence of sharedness index on individual well-being depending on the characteristics of individuals and teams. For example, we can assume an individual with a personality whose own well-being does not correlate with sharedness index. In addition, multiple sub-teams within a team may have different sharedness index and need to be stratified by sub-team. Furthermore, it may be important to consider whether the work is performed by a collaborative team or by individuals working independently.

To enhance individual well-being based on our findings, a qualitative exploration is necessary to identify team characteristics influencing the correlation between sharedness index and individual well-being. For example, Fukushima et al. [42] have shown that the strength of the correlation between individual well-being and neighborhood well-being varies with social capital. Knight et al. [43] have found that whether a group's negative emotions have a positive impact on task performance depends on whether the source of the emotions is outside or inside the group, and on the duration of the group. Tanghe et al. have shown that the degree of convergence of collective emotions is higher in groups with stronger member identification. As in these studies, it would be beneficial for us to investigate the intermediate features between the degree of well-being sharedness and individual well-being.

### Alternative explanations including self report coupling and sharedness

Also, the interpretation of correlations is controversial. The correlation coefficient between the mean of individual well-being and the standard deviation of individual well-being was 0.428, showing a higher positive correlation than that

between individual well-being and sharedness index. The correlation coefficient between individual well-being and team atmosphere is 0.815, which is very high. Based on these results, we can interpret the results of our correlation analysis in a different way from our hypothesis: participants tend to input values similar to their own well-being as their team atmosphere, therefore, participants are happier when their own well-being is consistent with each other's. This interpretation also seems to apply to the correlation between the text-based sharedness index and well-being. The diary can be interpreted as an expression of the psychological state of the individual as well as the perception of the team's state. Therefore, the correlation between the text-based sharedness index and well-being may indicate a correlation between the consistency of the psychological state within the team and the individual well-being. In other words, if everyone feels the same, everyone feels high well-being. In the future, to find out which story is appropriate, we can change our experimental setup from items about team atmosphere to items about other aspects of the team psychological state that is not related to well-being, or we could limit the collection of texts from diary to "perceptions about the team".

### Limited power in team and week strata (exploratory $p < 0.10$)

When analyzing the correlation of the data divided by team or by week, the cutoff for the p-value was set at 0.1. This is a 90% confidence interval, which is not a very high confidence interval. Therefore, to apply our results to improving well-being in the real world, it is essential to propose a comprehensive model that predicts well-being from the state of the group, using not only the degree of sharedness, but also complementary variables such as individual personality characteristics, team characteristics (such as whether they work together), and cohesion among members.

### Generalizability limits due to single company sample and weekly aggregation

Finally, because this experiment relied solely on data from a single company, it is plausible that factors unique to that specific company and the term of the experiment may have influenced the results. In future, we plan to conduct more long-term experiments with other companies. Furthermore, the number of data points for measuring sharedness index within the team was small, resulting in an analysis using weekly rather than daily data points. Therefore, increasing the number of participants per team and encouraging participant input is important in future experiments.

## Ethical considerations

This research was approved by the ethics committee of Kyoto University, including Nara Institute of Science and Technology (Review No. 26-P-16), and was conducted in accordance with relevant guidelines and regulations. All participants were fully informed about the study and provided consent prior to participation. Confidentiality and anonymity were assured, and no personally identifiable or company-confidential information was collected in the diaries. Participation was voluntary, and individuals could withdraw at any time without penalty. Participants and researchers belonged to different organizations, which minimized the risk of coercion or undue influence.

At the same time, ethical attention was required because the daily reporting task itself may have influenced participants' psychological states. Regularly reflecting on one's well-being could heighten self-awareness of mood, which might have both beneficial and burdensome effects. In addition, the feedback system displayed aggregated well-being levels of team members, which could have affected participants' feelings. For example, participants might have experienced distress if others in the team reported low well-being, or reassurance if others reported high well-being. Such potential influences on participants' emotions and team dynamics were carefully considered when designing the study. Future research should further explore ways to minimize unintended psychological impacts while preserving the benefits of self-reflection and collective feedback.

## Conclusion

To investigate the interpretable indicator of social harmony, this study examined the correlation between the degree of team psychological state shared and individual well-being based on the psychological theory of shared reality. The extent to which team psychological state was shared was measured using both a score-based method, which measured the agreement of team atmosphere scores among team members, and a text-based method, which measured semantic similarity in the diary texts. For both metrics, the results showed a positive correlation between individual well-being and sharedness index, supporting our hypothesis. Moreover, sharedness index was shown to be an alternative to the existing social harmony index. Future work will include analyses using linguistic features and the verification of causal relationships by setting experiments, including interventions. Additionally, it is important to investigate other types of team psychological states that affect individual well-being, as there are several possible team psychological states besides the degree of shared team atmosphere employed in this study, that may enhance team members' well-being and productivity.

## Supporting information

**S1 Table. All items on the Interdependent Happiness Scale [17].**
(DOCX)

**S2 Algorithm. Computation of $TSI_{t,w}$ for a team by week.**
(DOCX)

**S3 Table. Team-stratified correlations between sharedness indices and individual well-being.**
(DOCX)

**S4 Table. Weekly correlations between sharedness indices and individual well-being.**
(DOCX)

## Author contributions

**Conceptualization:** Masae Manabe, Masataka Nakayama, Yukiko Uchida, Eiji Aramaki.

**Data curation:** Kazuhiro Ito.

**Formal analysis:** Kazuhiro Ito.

**Investigation:** Kazuhiro Ito, Masae Manabe, Yasushi Watanabe.

**Methodology:** Kazuhiro Ito.

**Software:** Kazuhiro Ito.

**Supervision:** Eiji Aramaki.

**Visualization:** Kazuhiro Ito.

**Writing – original draft:** Kazuhiro Ito, Shoko Wakamiya.

**Writing – review & editing:** Kazuhiro Ito, Shoko Wakamiya.

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
