## [Decision Letter · Decision Letter 0]

7 Aug 2025

PONE-D-24-47890Team Atmosphere and Individual Well-Being: The Importance of Shared Perception in Japanese CompanyPLOS ONE

Dear Dr.  Aramaki, 

Thank you for submitting your manuscript to PLOS ONE. After careful consideration, we feel that it has merit but does not fully meet PLOS ONE’s publication criteria as it currently stands. Therefore, we invite you to submit a revised version of the manuscript that addresses the points raised during the review process. Please submit your revised manuscript by Sep 21 2025 11:59PM. If you will need more time than this to complete your revisions, please reply to this message or contact the journal office at plosone@plos.org. Please include the following items when submitting your revised manuscript:

We look forward to receiving your revised manuscript.

Kind regards,

Erhan Atay

Academic Editor

PLOS ONE

Journal Requirements:

 “This work was supported by JST-Mirai Program Grant Number JPMJMI21J2, Japan.” 

4. In the online submission form, you indicated that "The dataset collected and analysed during the current study are not publicly available as the dataset contains sensitive information regarding personal diaries and individual psychological states, thus precluding its public release but are available from the corresponding author on reasonable request."

“This work was supported by JST-Mirai Program Grant Number JPMJMI21J2, Japan.” 

“This work was supported by JST-Mirai Program Grant Number JPMJMI21J2, Japan.”

6. Please amend either the abstract on the online submission form (via Edit Submission) or the abstract in the manuscript so that they are identical.

7. Please ensure that you refer to Figure 1 in your text as, if accepted, production will need this reference to link the reader to the figure.

Additional Editor Comments 

Dear Authors,

Thank you for your submission, “Team Atmosphere and Individual Well-Being: The Importance of Shared Perception in Japanese Company.” Your work addresses an increasingly important topic in organizational psychology and offers a promising contribution through the innovative sharedness index. The two-month study is well-structured, and your dual approach using score-based and text-based metrics is compelling.

However, several revisions are necessary to strengthen the manuscript:

1. Title and Abstract Improvements:

The term “social harmony” is central to your research but is not clearly reflected in the title or consistently used in the abstract. We suggest rephrasing the title to better capture this concept, and restructuring the abstract to clearly articulate the study's aim, significance, methodology, and key findings (Reviewer 1 & 2).

2. Structural and Formatting Revisions:

Figures and tables should be referenced explicitly in the text, placed appropriately, and accompanied by concise captions. Currently, many figures (e.g., Fig 1, Fig 2, Fig 4) are inserted mid-paragraph without context, making the flow difficult to follow (Reviewer 1). Reconsider the location of Table 1 and revise Figure 7 caption errors (Reviewer 1).

3. Editing and Language Usage:

Substantive language editing is needed throughout the manuscript. Avoid using ampersands (&) outside of in-text citations and ensure consistency in reference formatting. Reviewer 1 also noted some issues with citation style in the references section, such as the incorrect handling of WHO as an author.

4. Missing Standard Sections:

Reviewer 2 highlighted missing or underdeveloped sections such as a clear research limitations section, details on ethical considerations, and a discussion on the study's budget or funding. Please expand on these areas to ensure the paper meets standard publication expectations.

5. Methodological Clarity:

Enhance the clarity of your methods section by consolidating overlapping content between “Materials” and “Methods,” and introducing a clearly defined “Data Analysis” subsection. Reviewer 2 also requested that you elaborate more on sample size, population details, and statistical methods used.

We encourage you to revise the manuscript with these points in mind. Your topic is highly relevant and shows promise, but substantial revisions are needed for clarity, coherence, and completeness.

We look forward to reviewing a revised version of your manuscript.

Sincerely,

Reviewer's Responses to Questions

**Comments to the Author**

1. Is the manuscript technically sound, and do the data support the conclusions?

Reviewer #1: Yes

Reviewer #2: Partly

2. Has the statistical analysis been performed appropriately and rigorously?

Reviewer #1: I Don't Know

Reviewer #2: No

3. Have the authors made all data underlying the findings in their manuscript fully available?

Reviewer #1: No

Reviewer #2: Yes

4. Is the manuscript presented in an intelligible fashion and written in standard English?

Reviewer #1: Yes

Reviewer #2: Yes

5. Review Comments to the Author

Reviewer #1: The paper is interesting and of value, yet the following are to be considered to enrich the work done:

1.I suggest re-phrasing the title to reflect the concept of ‘social harmony’ since it is the focus in the abstract. And also refer to it in the body of the paper.

2.Editing is required.

3.The abstract needs further improvement since the study aim, gap, significance are not clearly introduced. Besides, the model of analysis should be documented.

4.Figure 1 appears all of a sudden when the paragraph hasn’t come to an end. Besides, there is no reference to it, before or after.

5.Table one appears in the middle of the paragraph on ‘Participant’ with no reference to it after or before it. It is not in the right place. It should come as part of the Questionnaire section.

6.Figure 2 should be situated after the first paragraph of the Instructions section.

7.It is better to merge both sections Materials and Methods since both deal with the methodological consideration or the Methodology of the paper.

8.The Analysis part of the Methods section should be separated and titled ‘Analysis’ or ‘Data Analysis’.

9.No reference to Figure 3 is seen in the main text.

10.I suggest having an introductory statement or account before Figure 4 to prepare the reader to what comes after. Besides, the caption of the figure is too long. All figures are followed by very long captions. Reduce, please.

11.When explaining in the section ‘text-based sharedness index’, in the second paragraph, Fig 7 is mentioned while this section is followed by Fig. 6. Mismatch is there in different places.

12.The ampersand ‘&’ should not be used within the text (as in line 372 and some other places), only used within the ‘in-text citation’.

13.Figures and tables should be referred to in the main text immediately after or before them.

14.In the third reference in the Reference list, spell the whole words of WHO rather than treating the title of the organization as an authors’ name.

15.References in the Reference List need double checking for some minor issues.

Best of luck.

Reviewer #2: Dear Authors,

The following suggestions may help improve your article:

1. In a structured abstract, it can include a statement that states the main purpose of the research. It should be included only in the introduction of topics that do not require references. Method: It should be stated precisely, including the study population, sample size, analysis methods, etc. The findings should be stated by stating the most important topic and based on statistical analysis, the topics should be raised. If each hypothesis is meaningful, state it by stating its significance. Conclusion: It should be expressed based on the study findings with a more general approach. 2. Introduction:

Adhere to a more precise structure. By stating the importance of the topic and addressing it in the world and the Asian region, address the growing or declining trend of this topic and by examining this topic and defining the relationship between variables and its impact role in Japanese companies and comparing it with a systematic approach, discuss the rationale and explanation of the why and philosophy of your research. The article that is used as a research method should be included in the method and methods section.

3. Method: By designing a model of the whole except in detail about the implementation method, statistical population and formula in a title, how to collect data, tools or data on official sites, how to process and finally from statistical software and methods.

4. All the results of data analysis in the findings section should be presented in tables starting with the most important findings for better summary and understanding.

5. Regarding ethics in research in this research, it should be mentioned

6. In the area of budget of this research, it should be mentioned

7. Research limitations should be mentioned in detail as a title

In general, the precise and intelligent title of this article can help with the mentioned amendments in a more accurate presentation of those interested in this topic

6. PLOS authors have the option to publish the peer review history of their article (what does this mean?). If published, this will include your full peer review and any attached files.

Reviewer #1: **Yes: **Nawal Fadhil Abbas

Reviewer #2: No

---

## [Author Response · Author response to Decision Letter 1]

1 Oct 2025

Please see the attached file "Response to Reviewers" for our detailed responses to each reviewer and editor comment.

We have addressed all points raised in the decision letter and revised the manuscript accordingly.

---

## [Decision Letter · Decision Letter 1]

1 Dec 2025

PONE-D-24-47890R1Social harmony at work: A sharedness index linking team atmosphere to individual well being in a Japanese companyPLOS ONE

Dear Dr. Aramaki,

Thank you for submitting your manuscript to PLOS ONE. After careful consideration, we feel that it has merit but does not fully meet PLOS ONE’s publication criteria as it currently stands. Therefore, we invite you to submit a revised version of the manuscript that addresses the points raised during the review process.

We look forward to receiving your revised manuscript.

Kind regards,

Gal Harpaz, Ph.D.

Academic Editor

PLOS ONE

Journal Requirements:

**Additional Editor Comments:**

Dear Authors,

Thank you for submitting the revised version of your manuscript to PLOS ONE. The reviewers have now evaluated your revision and are satisfied that you have addressed their comments thoroughly and effectively.

Based on the positive feedback from both reviewers and my own assessment of the revised manuscript, I am pleased to inform you that your paper is accepted for publication in PLOS ONE.

Congratulations to you and your co-authors on this accomplishment, and thank you for choosing PLOS ONE as the venue for your work.

With best regards,

Dr. Gal Harpaz

Academic Editor

PLOS ONE

Reviewers' comments:

Reviewer's Responses to Questions

**Comments to the Author**

1. If the authors have adequately addressed your comments raised in a previous round of review and you feel that this manuscript is now acceptable for publication, you may indicate that here to bypass the “Comments to the Author” section, enter your conflict of interest statement in the “Confidential to Editor” section, and submit your "Accept" recommendation.

Reviewer #2: All comments have been addressed

Reviewer #3: All comments have been addressed

Reviewer #4: (No Response)

2. Is the manuscript technically sound, and do the data support the conclusions?

Reviewer #2: Yes

Reviewer #3: Yes

Reviewer #4: Yes

3. Has the statistical analysis been performed appropriately and rigorously?

Reviewer #2: Yes

Reviewer #3: I Don't Know

Reviewer #4: Yes

4. Have the authors made all data underlying the findings in their manuscript fully available?

Reviewer #2: Yes

Reviewer #3: Yes

Reviewer #4: Yes

5. Is the manuscript presented in an intelligible fashion and written in standard English?

Reviewer #2: Yes

Reviewer #3: Yes

Reviewer #4: Yes

6. Review Comments to the Author

Reviewer #2: (No Response)

Reviewer #3: Dear Authors,

I recommend your manuscript titled “Social harmony at work: A sharedness index linking team atmosphere to individual well-being in a Japanese company” for publication in its current form.

Congratulations! Wishing you continued success.

Reviewer #4: Thank you for a manuscript which describes your data collection, analysis and interpretations of the work. The manuscript doesn't quite follow a succinct paper structure, but does indicate what was done.

I have some particular comments:

line 274: there is a missing symbol to indicate < for the p-value, and also in the caption for Figure 4 the symbol is not correct.

lines 305 and 328 have <= and the captions for Fig 5 and 7 have < alone - this should be consistent.

7. PLOS authors have the option to publish the peer review history of their article (what does this mean?). If published, this will include your full peer review and any attached files.

Reviewer #2: No

Reviewer #3: No

Reviewer #4: No

---

## [Author Response · Author response to Decision Letter 2]

4 Dec 2025

We have uploaded a detailed point-by-point response to the reviewers' comments as a separate file labeled "Response to Reviewers". In this revision, we have addressed the technical corrections regarding statistical symbols pointed out by Reviewer #4 and ensured consistency throughout the manuscript.

---

## [Editor Report · Decision Letter 2]

9 Dec 2025

Social harmony at work: A sharedness index linking team atmosphere to individual well being in a Japanese company

PONE-D-24-47890R2

Dear Dr. Aramaki,

We’re pleased to inform you that your manuscript has been judged scientifically suitable for publication and will be formally accepted for publication once it meets all outstanding technical requirements.

Kind regards,

Gal Harpaz, Ph.D.

Academic Editor

PLOS One

---

## [Editor Report · Acceptance letter]

PONE-D-24-47890R2

PLOS One

Dear Dr. Aramaki,

I'm pleased to inform you that your manuscript has been deemed suitable for publication in PLOS One. Congratulations! Your manuscript is now being handed over to our production team.

Kind regards,

on behalf of

Dr. Gal Harpaz

Academic Editor

PLOS One